# Effect of *Aspergillus flavus* Fungal Elicitor on the Production of Terpenoid Indole Alkaloids in *Catharanthus roseus* Cambial Meristematic Cells

**DOI:** 10.3390/molecules23123276

**Published:** 2018-12-11

**Authors:** Chuxin Liang, Chang Chen, Pengfei Zhou, Lv Xu, Jianhua Zhu, Jincai Liang, Jiachen Zi, Rongmin Yu

**Affiliations:** 1Biotechnological Institute of Chinese Materia Medica, Jinan University, 601 Huangpu Avenue West, Guangzhou 510632, China; lcxliangcx@163.com (C.L.); xulv1818@163.com (L.X.); 2Department of Natural Product Chemistry, College of Pharmacy, Jinan University, Guangzhou 510632, China; imchenchang@163.com (C.C.); yffsljc2006@sina.com (J.L.); jiachen_zi@163.com (J.Z.); 3Department of Basic Medical Sciences, Xinxiang Medical University, Xinxiang 453003, China; zpf861223@163.com

**Keywords:** *Catharanthus roseus*, cambial meristematic cells, *Aspergillus flavus*, terpenoid indole alkaloids, biosynthesis

## Abstract

This study reported the inducing effect of *Aspergillus flavus* fungal elicitor on biosynthesis of terpenoid indole alkaloids (TIAs) in *Catharanthus roseus* cambial meristematic cells (CMCs) and its inducing mechanism. According to the results determined by HPLC and HPLC-MS/MS, the optimal condition of the *A. flavus* elicitor was as follows: after suspension culture of *C. roseus* CMCs for 6 day, 25 mg/L *A. flavus* mycelium elicitor were added, and the CMC suspensions were further cultured for another 48 h. In this condition, the contents of vindoline, catharanthine, and ajmaline were 1.45-, 3.29-, and 2.14-times as high as those of the control group, respectively. Transcriptome analysis showed that *D4H*, *G10H*, *GES*, *IRS*, *LAMT*, *SGD*, *STR*, *TDC*, and *ORCA3* were involved in the regulation of this induction process. The results of qRT-PCR indicated that the increasing accumulations of vindoline, catharanthine, and ajmaline in *C. roseus* CMCs were correlated with the increasing expression of the above genes. Therefore, *A. flavus* fungal elicitor could enhance the TIA production of *C. roseus* CMCs, which might be used as an alternative biotechnological resource for obtaining bioactive alkaloids.

## 1. Introduction

*Catharanthus roseus* (L.) Don is a perennial medicinal plant of the family Apocynaceae. At present, over 130 alkaloids have been isolated from *C. roseus*, most of which are terpenoid indole alkaloids (TIAs) [1]. Some TIAs such as vinblastine, vincristine, and ajmaline exhibit strong pharmacological activities, and some are widely used in the treatment of various diseases [2,3,4,5]. Ajmalicine is a potent antihypertensive reagent [5]. Vinblastine and vincristine, two bisindole alkaloids derived from coupling vindoline and catharanthine, are natural anticancer drugs and are still among the most valuable agents used to treat cancer [3,4].

These secondary metabolites are produced from the TIA biosynthetic pathway in *C. roseus*, which is complex and highly regulated (Figure 1) [2,6,7]. The TIA biosynthetic pathway consists of TIA feeder pathways and the downstream of the TIA biosynthetic pathway. The TIA feeder pathways are the monoterpenoid pathway and indole pathway [7]. The downstream of the TIA biosynthetic pathway in *C. roseus* starts with the formation of strictosidine from tryptamine and secologanin, which is catalyzed by strictosidine synthase (STR) [8]. Then, strictosidine is deglucosylated by strictosidine β-d-glucosidase (SGD) to form strictosidine aglycone. Further enzymatic steps result in the formation of numerous TIAs, and the TIA biosynthetic pathway is classified into several specific branches. One branch of the TIA biosynthetic pathway produces ajmalicine and serpentine, a second branch catharanthine, a third vindoline, and a fourth lochnericine and horhammericine [7]. Genes encoding enzymes catalyzing the production of vindoline from tabersonine, such as deacetoxyvindoline 4-hydroxylase (D4H) and deacetylvindoline O-acyltransferase (DAT), have now been identified [9,10,11,12,13,14]. The formation of α-3′,4′-anhydrovinblastine from vindoline and catharanthine is catalyzed by a major class III peroxidase (PRX1) [15]. Then, vinblastine and vincristine are formed through multiple enzymatic steps from α-3′,4′-anhydrovinblastine (Figure 1) [7]. The TIA biosynthetic pathway in *C. roseus* is highly regulated by enzymes and transcription factors (TFs). In 2000, Fits and Memelink discovered octadecanoid-derivative responsive *Catharanthus* A P2-domain protein 3 (ORCA3), a jasmonate-responsive AP2/ERF transcription factor, in *C. roseus* using the T-DNA activation tagging technology [16]. Since overexpression of *ORCA3* increases the expression of key genes in the TIA biosynthetic pathway, it is considered to be the core transcription factor in the TIA biosynthetic pathway in *C. roseus* [6]. In addition, *ORCA3* expression could be induced by jasmonates [17]. 

However, the content of the pharmacological secondary metabolites in *C. roseus* is very low [18], and commercial production has used a semi-synthetic route to couple catharanthine and vindoline [3]. Therefore, developing methods to increase the yields of TIAs in *C. roseus* has become a focus for domestic and foreign scholars [19]. In recent years, an increasing number of studies has shown that fungal elicitors play an important role in regulating biological secondary metabolic pathways and intracellular information transmission [20,21]. The fungal elicitor is an active substance derived from fungus, which can rapidly and specifically induce the expression of specific plant genes, thereby activating specific secondary metabolic pathways and increasing the accumulation of interesting secondary metabolites [20]. The treatment of plant cell culture systems with fungal elicitors has become an effective method for rapidly increasing the yield of target product in plant cell culture [20,21]. Although fungal elicitors are known to regulate the production of secondary metabolites, the inducting effect varies depending on factors such as fungal species, elicitor components, dose, time of addition, and type of plant cell culture [22,23,24]. In 2016, Tonk reported that a low-dose *A. flavus* fungal elicitor effectively increased the growth rate of callus, embryo biomass, germination rate, and alkaloid content in the embryos of *C. roseus*, as well as increasing the shoot and root lengths of germinated somatic embryos. Further, antioxidant enzyme activity assays showed that a low-dose *A. flavus* elicitor caused an allergic reaction in the mature and germinating somatic embryos of *C. roseus*, resulting in an increase in alkaloid content [24]. Therefore, *A. flavus* may be a good elicitor for promoting the TIA production of *C. roseus* cell cultures.

More interestingly, our previous research showed that *C. roseus* cambial meristematic cells (CMCs) exhibit good characteristics when compared to *C. roseus* dedifferentiated plant cells (DDCs), such as faster growth, higher yields, diverse bioactive TIAs, lower variability, and high expression of TIA biosynthesis genes. In addition, after two years of being cultured, *C. roseus* CMCs remained stable in both genetic traits and alkaloid content; the production of vindoline, catharanthine, and ajmaline in *C. roseus* CMCs could be enhanced by β-cyclodextrin and methyl jasmonate (MeJA) [25]. Therefore, *C. roseus* CMCs may be a good system for the investigation of the biosynthesis and regulation of TIAs. 

However, there is no report on the effect of the *A. flavus* fungal elicitor on the TIA production of *C. roseus* CMCs so far. In this research, we investigated the inducing effect of the *A. flavus* fungal elicitor on the biosynthesis of TIAs in *C. roseus* CMCs, and the inducing mechanism was explored by transcriptome analysis and determination of the expression of TIA biosynthesis-related genes via the quantitative real-time reverse transcription polymerase chain reaction (qRT-PCR) technique.

## 2. Results

### 2.1. Dry Cell Weight in Response to A. flavus Elicitation

Different concentrations of *A. flavus* medium elicitor were added to four-day-old or six-day-old suspensions of *C. roseus* CMCs. In the four-day-old suspensions of CMCs, we observed that the dry cell weight was slightly higher than that of the check group (CK) after *A. flavus* medium elicitor treatment for 24 h (Figure 2a). In the six-day-old suspensions of CMCs, we observed no significant difference in dry cell weight after the addition of the *A. flavus* medium elicitor (Figure 2b). Besides, the concentration of the *A. flavus* mycelium elicitor had no significant effect on the dry cell weight of *C. roseus* CMC suspension cultures (Figure 2c,d). For six-day-old suspensions of CMCs, the dry cell weight was slightly higher than that of the CK after 24-h treatment of 15 mg/L *A. flavus* mycelium elicitor (Figure 2d). Based on the results of Figure 2, the *A. flavus* elicitor (different concentrations: 5, 15, or 25 mg/L) had no negative effect on the growth of *C. roseus* CMCs in the selected concentration.

### 2.2. HPLC-MS/MS Analysis of Alkaloids 

The *C. roseus* CMC cultures were harvested after suspension culture for six days. Then, the separation of the compounds of *C. roseus* CMCs was performed by reversed-phase high-performance liquid chromatography (RP-HPLC) with photodiode array detection, as well as by electrospray ionization tandem mass spectrometry (ESI-MS/MS) in positive mode (Appendix A). The MS spectra of the identified compounds are displayed in Appendix A.

Compound **1**, identified by MS and chromatographic behavior comparison with that of the authentic standard, was ajmalicine (Rt 26.9 min. +MS: 353 [M + H]^+^; +MS2: 353, 321, 284, 252, 222, 210, 178, 144, 143, 117) (Appendix A).

Compound **2**, equally identified by MS and chromatographic behavior comparison with that of the authentic standard, was catharanthine (Rt 31.3 min. +MS: 337 [M + H]^+^; +MS2: 337, 248, 219, 204, 173, 165, 144, 143, 133, 128, 127, 93, 91, 77) (Appendix A).

Another compound (**3**) with [M + H]^+^ at 457 putatively corresponded to vindoline. Compound **3** (Rt 37.1 min) with [M + H]^+^ at 337 was noticed, and its UV spectrum (UV: 240, 294 nm) was similar to that of catharanthine; therefore, it was the catharanthine isomer. Further comparison showed that its MS spectrum (+MS: 457 [M + H]^+^; +MS2: 457, 439, 397, 379, 347, 337, 295, 258, 232, 222, 188, 173, 162, 157, 145, 134, 122) (Appendix A) was in line with that of the vindoline authentic standard. 

### 2.3. TIA Content in Response to Elicitation

To obtain the optimal *A. flavus* elicitor treatment condition, different concentrations of the *A. flavus* medium elicitor and mycelium elicitor were tested for their inducing effects in *C. roseus* CMC suspension cultures.

The effect of the *A. flavus* elicitor on TIA content in *C. roseus* CMCs is shown in Figure 3 and Figure 4. As shown in these figures, *A. flavus* increased the TIA content in *C. roseus* CMCs. For different alkaloids, the induction condition for the highest yield was different. The content of catharanthine reached its maximum (3.39 mg/L), which was 3.6-times as high as that of the CK, after 24-h treatment with 25 mg/L *A. flavus* medium elicitor in four-day-old suspensions of CMCs (Figure 3b). The content of vindoline reached as high as 8.79 mg/L, which was 1.45-times as high as that of the CK, after 48-h treatment with 25 mg/L *A. flavus* mycelium elicitor in six-day-old suspensions of CMCs (Figure 4a). As for ajmaline, the content reached 9.84 mg/L, which was 3.4-times as high as that of CK, after 24-h treatment with 25 mg/L *A. flavus* mycelium elicitor in six-day-old suspensions of CMCs (Figure 4c).

According to the timing when the contents of vindoline, catharanthine, and ajmaline were all relatively high (compared with those of the CK and other experimental groups), the optimal condition was confirmed. The optimal condition for the *A. flavus* elicitor treatment in *C. roseus* CMCs was as follows: after suspension culture of *C. roseus* CMCs for six days, 25 mg/L *A. flavus* mycelium elicitor were added, and the CMC suspensions were further cultured another 48 h. Although the content of total alkaloids was slightly lower than that of the CK under this condition, the contents of vindoline, catharanthine, and ajmaline reached 8.79, 2.81, and 8.95 mg/L, respectively, which were 1.45-, 3.29-, and 2.14-times as high as those of the CK, respectively (Figure 4e–g).

### 2.4. Functional Annotation and Functional Classification of Unigenes

To identify the key genes in the TIA biosynthetic pathway in *C. roseus* CMCs, the transcriptomes of *C. roseus* DDCs (callus) treated with an equal amount of sterile water, *C. roseus* CMCs (the check treated with an equal amount of sterile water, CK), and *C. roseus* CMCs under optimal *A. flavus* elicitor treatment condition (the experimental group, EG) were determined.

In total, approximately 170.7 million Illumina raw data were generated from the three different samples (Appendix A). After filtering the raw data, approximately 59.7, 50.7, and 57.8 million clean reads remained for the callus, CK, and EG transcriptomes, respectively (Appendix A). All clean reads were subsequently subjected to de novo assembly with the Trinity program, producing 121,532 transcripts and 105,552 unigenes (Appendix A). 

Functional annotations of unigenes in the seven largest databases are shown in Appendix A. A total of 83,742 unigenes (79.33%) were annotated based on the information available from seven public protein databases including the NCBI non-redundant protein sequences (NR), NCBI non-redundant nucleotide sequences (NT), Swiss-Prot, Protein family (Pfam), Gene Ontology (GO), euKaryotic Ortholog Groups (KOG), and the Kyoto Encyclopedia of Genes and Genomes (KEGG) using the Basic Local Alignment Search Tool (BLAST) with an *E*-value cut-off of 1 e^−10^ (Appendix A). A total of 79,711 unigenes (75.51% of the total assembled unigenes) had a match in the NR database, and 59,272 (56.15%), 63,496 (60.15%), 61,591 (58.35%), 61,829 (58.57%), 26,674 (25.27), and 34,367 (32.55%) unigenes showed significant similarity to sequences in the NT, Swiss-Prot, Pfam, GO, KOG, and KEGG databases, respectively (Appendix A). 

All unigenes were subjected to a search against the GO database to classify unigene functions based on the NR annotation. Of the 105,552 assembled unigenes, 61,829 unigenes were successfully assigned to one or more GO terms, and these unigenes were classified into three main GO categories and 56 groups (Appendix A). Within the “biological process” domain, the most evident matches were the terms “cellular process” (37,633), “metabolic process” (35,132), and “single-organism process” (28,250). In the “cellular component” domain, the terms “cell” (20,343) and “cell part” (20,343) were most frequently assigned. For the “molecular function” domain, the assignments were mostly enriched in the terms “binding” (37,680) and “catalytic activity” (31,249). 

For further analysis, the unigenes were mapped onto the KEGG database for categorization of gene function and identification of biochemical pathways. A total of 34,367 unigenes were annotated and assigned to 5 main KEGG metabolic pathways, 19 sub-branches, and 300 KEGG pathways. Among them, the most common sub-branch was “carbohydrate metabolism” (2904), followed by “translation” (2711) and “folding, sorting and degradation” (2575) (Appendix A). In addition, there were seven unigene matches in “indole alkaloid biosynthesis” (ko00901), 43 in “monoterpenoid biosynthesis” (ko00902), and 314 in “terpenoid backbone biosynthesis” (ko00900) (Appendix A). 

As shown in Appendix A, the detected unigenes contained transcription factors of the orphans, AP2-EREBP, and SET families.

### 2.5. High and Differential Expression Analysis of Unigenes

The correlation of gene transcription levels between samples could reflect differences in gene expression patterns. The closer the correlation coefficient is to one, the higher the degree of similarity of the gene expression pattern between samples. On the other hand, the closer the correlation coefficient is to zero, the bigger the difference in gene expression pattern between samples. As shown in Appendix A, the square of the Pearson correlation coefficient (R^2^) was 0.539, indicating that there was a significant difference between the callus and CK groups in the level of gene expression. Thus, the gene expression pattern of the *C. roseus* CMCs was very different from that of the *C. roseus* DDCs. Besides, R^2^ was 0.86 between the EG and CK groups (Appendix A), showing that *A. flavus* elicitor treatment could indeed cause differences in gene expression levels.

Differentially-expressed genes (DEGs) (|log_2_ (fold change)| ≥ 1 and *q*-value ≤ 0.005) were defined as unigenes that were significantly enriched or depleted in one sample relative to the other. A volcano plot was constructed to illustrate the distribution of DEGs in the callus vs. CK and EG vs. CK groups (Appendix A). The results of the differential expression analysis indicated that the expression levels of some genes in the callus group were significantly up- or down-regulated when compared with the CK. Further, compared with the CK, the expression levels of a few genes in the EG group were significantly up- or down-regulated, but the significant degree was lower than that of the callus vs. CK. The number of common differential genes among the two comparative combinations was 133, while the number of unique differential genes was 4825 in callus vs. CK and 139 in EG vs. CK (Appendix A).

To further understand the biological functions of DEGs, they were annotated with the GO and KEGG pathways. Compared with the CK, there were 2035 up-regulated DEGs and 1859 down-regulated DEGs in the callus group and 232 up-regulated DEGs and 114 down-regulated DEGs in EG (Appendix A). Partial results of KEGG pathways analysis of DEGs, which were associated with TIA biosynthesis and induced by the *A. flavus* elicitor, are shown in Appendix A. Compared with the *C. roseus* DDCs (the callus group), some genes related to the biosynthesis of TIAs were up-regulated in the *C. roseus* CMCs (the CK group). Combining the results of differential expression analysis, GO enrichment, and KEGG pathway analysis of DEGs in callus vs. CK and EG vs. CK, the DEGs associated with TIA biosynthesis and induced by the *A. flavus* elicitor were screened, and their sequences were aligned with the genes in the NCBI database via the BLAST tool. The genes identified were *D4H*, 10-hydroxylase geraniol (*G10H*), geraniol synthase (*GES*), iridoid synthase (*IRS*), loganic acid O-methyltransferase (*LAMT*), *SGD*, *STR*, tryptophan decarboxylase (*TDC*), and *ORCA3*, of which *ORCA3* was a transcription factor gene.

### 2.6. qRT-PCR

Under the optimal *A. flavus* elicitor treatment condition, the transcription levels of *D4H*, *G10H*, *GES*, *IRS*, *LAMT*, *SGD*, *STR*, *TDC*, and *ORCA3* were much higher in EG than in CK. Specifically, their expression were 4.49-, 1.75-, 1.71-, 1.42-, 3.12-, 2.33-, 2.87-, 2.51-, and 5.97-times as high as those of CK, respectively (Figure 5). 

## 3. Discussion

Since the low content of the pharmacological TIAs in *C. roseus* [18] and commercial production has used a semi-synthetic route to couple vindoline and catharanthine [3], developing methods to enhance the TIA production of *C. roseus* cell cultures has become a focus for domestic and foreign scholars [19]. According to our previous research, undifferentiated *C. roseus* CMCs were capable of maintaining not only good cellular morphology, but also stable and high production of alkaloid metabolites [25]. Therefore, *C. roseus* CMCs were used as plant cell materials for the investigation of the biosynthesis of TIAs. 

In previous reports, fungal elicitors could increase the accumulation of secondary metabolites of interest by activating specific secondary metabolic pathways [20]. Tonk et al. showed that not only the callus growth rate, but also the alkaloid content in the embryos of *C. roseus* could be effectively increased after being treated with a low-dose *A. flavus* fungal elicitor [24]. However, there has been no research on the effect of *A. flavus* fungal elicitor on the yields of TIAs in *C. roseus* CMCs. Generally, the optimal time for adding the fungal elicitor to cell cultures is at the log phase of the culture cycle, when the cell cultures are most sensitive to elicitation. Our previous research showed that the log phase of the *C. roseus* CMC culture cycle lasts from 4–9 days [25]. Besides, the results of our preliminary experiment in this research showed that the *C. roseus* CMC cultures turned brown after three days of treatment or when the suspension cultures of *C. roseus* CMCs were treated with the fungal elicitor after a nine-day culture. Thus, we finally added three different concentrations of the *A. flavus* medium elicitor or the *A. flavus* mycelium elicitor to four-day-old or six-day-old suspensions of *C. roseus* CMCs. After inducing treatment, we found that the *A. flavus* mycelium elicitor could promote the TIA production without a negative effect on the growth of *C. roseus* CMCs in the selected concentration. According to the results determined by HPLC and HPLC-MS, the optimal condition of the *A. flavus* elicitor was as follows: after suspension culture of *C. roseus* CMCs for six days, 25 mg/L *A. flavus* mycelium elicitor were added, and the CMC suspensions were further cultured for another 48 h. Under this condition, the contents of vindoline, catharanthine, and ajmaline were 1.45-, 3.29-, and 2.14-times as high as those of the CK group, respectively (Figure 4e–g). 

The TIA biosynthetic pathway in *C. roseus* is complex and highly regulated [2,6,7,8]. Hao et al. comprehensively analyzed the differential gene expression profiles of 24 h of continuous light, 24 h of dark treatment, 4 h of MeJA treatment under continuous light conditions, and 4 h of MeJA treatment under dark conditions in *Artemisia annua* seedlings using Illumina transcriptome sequencing. As a result, some TFs in the light signaling pathway were identified that can respond to MeJA [26]. Thus, transcriptome analysis is an effective method of investigating the inducing mechanism. To explore the inducing mechanism at the gene expression level, we analyzed the transcriptome data of *C. roseus* DDCs treated with sterile water, *C. roseus* CMCs treated with sterile water, and *C. roseus* CMCs under the optimal *A. flavus* elicitor treatment condition. This transcriptome analysis showed that *D4H*, *G10H*, *GES*, *IRS*, *LAMT*, *SGD*, *STR*, *TDC*, and *ORCA3* were involved in the regulation of this induction process. The functions of the above genes in the TIA biosynthetic pathway are shown in Figure 1. Among the above genes, *ORCA3* is the core transcription factor gene in the TIA biosynthetic pathway [6]. Then, we further analyzed the transcription levels of the above genes by qRT-PCR. The results of qRT-PCR showed that, under the optimal *A. flavus* elicitor treatment condition, the transcription levels of *D4H*, *G10H*, *GES*, *IRS*, *LAMT*, *SGD*, *STR*, *TDC*, and *ORCA3* were much higher in EG than in CK (Figure 5). These results indicated that the increasing accumulations of vindoline, catharanthine, and ajmaline in *C. roseus* CMCs were correlated with the increasing expression of the above genes. As previous reports showed, the expression of the transcription factor *ORCA3* gene could be induced by elicitors (such as MeJA and jasmonic acid), and elicitors could increase the expression of *D4H*, *G10H*, *STR*, *GES*, *SGD*, and *TDC* [6,17,25,27,28]. Since overexpression of *ORCA3* increases the expression of key genes such as *STR* in the TIA biosynthetic pathway [6], it is extremely meaningful to explore whether the up-regulation of TIA biosynthesis-related genes is related to the increasing transcription level of *ORCA3* under the *A. flavus* mycelium elicitor treatment. 

In conclusion, the *A. flavus* mycelium elicitor could promote the TIA production of *C. roseus* CMCs. Under the optimal *A. flavus* elicitor treatment condition, the contents of vindoline, catharanthine, and ajmaline were 1.45-, 3.29-, and 2.14-times as high as those of the CK group, respectively. Transcriptome analysis and qRT-PCR experiment revealed that *D4H*, *G10H*, *GES*, *IRS*, *LAMT*, *SGD*, *STR*, *TDC*, and *ORCA3* were involved in the regulation of this induction process. Furthermore, the up-regulation of TIA biosynthesis-related genes was related to the inducting effect of the *A. flavus* mycelium elicitor, which in turn promoted the accumulation of vindoline, catharanthine, and ajmaline in *C. roseus* CMCs. 

## 4. Materials and Methods 

### 4.1. Plant Materials and Culture Conditions

*C. roseus* CMCs were established using the reported method [25], and *C. roseus* DDCs were established using the method of [29]. *C. roseus* CMCs were placed on pH-adjusted (5.75–5.80) Murashige and Skoog (MS) solid medium [30] supplemented with 2.0 mg/L α-naphthalene acetic acid (NAA), 10 g/L sucrose, and 4 g/L gellan gum, cultured at 25 °C in darkness, and maintained by serial subculturing every 12 days [25]. *C. roseus* DDCs were sub-cultured under the same condition as that of *C. roseus* CMCs.

Four or six days prior to the fungal elicitor experiments, CMCs were inoculated into 100 mL of MS liquid medium supplemented with 2.0 mg/L NAA and 20 g/L sucrose in a 250-mL Erlenmeyer flask at concentrations of 50.0 g fresh weight/L [25] and grown at 25 °C at 110 rpm with a 24-h light photoperiod. The suspension culture condition of *C. roseus* DDCs was the same as that of *C. roseus* CMCs.

### 4.2. Procurement, Fungal Culture, and Preparation of Elicitor

*A. flavus* elicitors were prepared using the methods of Tonk et al. [24] with slight modifications. The *A. flavus* strain (CGMCC No. 3.6434) was obtained from the China General Microbiological Culture Collection Center (CGMCC, Beijing, China). The fungus was grown at 28 °C in 100-mL Erlenmeyer flasks containing potato dextrose agar (BAM Media M127, Guangdong Huankai Microbial Sci.&Tech.Co., Ltd., Guangzhou, China) and maintained by serial subculture every 7 days. After 7 days of subculture, the mycelium was separated from the fungal mat with forceps. 

The mycelium was washed several times with distilled water and homogenized with distilled water at 5.0, 15.0, and 25.0 g/L. Then, *A. flavus* mycelium elicitors were sterilized at 120 °C for 20 min and stored at 4 °C. At the same time, *A. flavus* medium elicitors were prepared from the fungal mat by the same method used to prepare *A. flavus* mycelium elicitors and stored at 4 °C.

### 4.3. Induction Treatment with Fungal Elicitor

The *A. flavus* fungal elicitor (0.1 mL) (*A. flavus* mycelium elicitor or *A. flavus* medium elicitor) was added to 4-day-old or 6-day-old suspensions of CMCs at final concentrations of 5, 15, and 25 mg/L. The check group (CK) was treated with 0.1 mL sterile distilled water. Then, the CMCs were further cultured at 25 °C at 110 rpm with a 24-h light photoperiod. After culturing for another 12, 24, 36, 48, and 60 h, the cultures and liquid media were harvested.

### 4.4. Measurement of Dry Cell Weight

The *C. roseus* CMCs were separated from the media by vacuum filtration and transferred to a Petri dish lined with a piece of filter paper. The cells were dried to constant weight at 50 °C and weighed on a ten-thousandth scale. All samples were measured in triplicate.

### 4.5. Alkaloid Extraction and Determination

#### 4.5.1. Chemicals

Vindoline, catharanthine, and ajmalicine were HPLC grade and were purchased from Aladdin (Aladdin Reagents Co., Shanghai, China), and all other chemicals were of analytical grade.

#### 4.5.2. HPLC-MS Analysis of Alkaloids

The alkaloid extraction from 6-day-old suspensions of *C. roseus* CMCs was performed according to the methods of Wang [31]. Alkaloid extracts were dissolved in 1.0 mL of methanol and analyzed by HPLC-MS/MS according to the methods of Ferreres et al. [32] with slight modifications.

The HPLC system was equipped with an Agilent 1260 series system (Agilent Technologies, Santa Clara, California, USA). Chromatographic separations were carried out on a 250 mm × 4.6 mm, 5 μm, Phenomenex Gemini C18 column (Phenomenex Inc., Torrance, CA, USA) at 25 °C. Elution was performed with a flow rate of 1 mL/min. Solvents used were acetonitrile (A) and acetic acid 1% (B). The gradient was as follows: 12% A at 0 min, 20% A at 30 min, 50% A at 40 min, 50% A at 45 min, 12% A at 47 min, 12% A at 55 min and the injection volume 10 μL. Spectroscopic data from all peaks were accumulated in the range 240–400 nm, and chromatograms were recorded at 255, 280, 290, 306, and 320 nm.

The mass detector was an X500R QTOF mass spectrometer (AB Sciex Pte. Ltd., Redwood City, CA, USA) equipped with an electrospray ionization (ESI) system and controlled by SCIEX OS 1.3.1 software (AB Sciex Pte. Ltd., Redwood City, CA, USA). The experimental parameters were as follows: curtain gas, 30 psi; ion source gas, GS1 60 psi and GS2 60 psi; temperature, 550 °C; ion spray voltage, 5.5 kV; collision gas, 7 psi. Other parameters were as follows: declustering potential (DP), 80 V; collision energy (collision energy spread), 10 V (0 V), and 35 V (15 V). The mass spectrometer was operated in the full-scan mode in the *m*/*z* range of 100–1500. The information-dependent acquisition spectra were automatically performed with helium as the collision gas in the *m*/*z* range of 50–1500. MS data were acquired in positive ionization mode. The isolation width of the parent ions for the following MS fragmentation events was set at ±50 mDa.

#### 4.5.3. Quantitative Analysis of Alkaloids

Alkaloid extraction was performed according to the methods of Wang et al. [31]. Alkaloid extracts were dissolved in 1.0 mL of methanol and analyzed by HPLC and ion-pair extraction-spectrophotometry as described by Yang et al. [33].

The contents of vindoline, catharanthine, and ajmalicine were determined at 25 °C by HPLC analysis using an Agilent 1260 series system (Agilent Technologies, Santa Clara, CA, USA) and a Phenomenex Gemini C18 column (250 mm × 4.6 mm, 5 μm) (Phenomenex Inc., Torrance, CA, USA). The mobile phase consisted of methanol/acetonitrile/10 mM ammonium acetate (15:40:45, *v*/*v*/*v*) at a flow rate of 1.0 mL/min. The detection wavelength was 280 nm, and the injection volume was 10 μL. Before injection, all samples were filtered with 0.45-μm nylon membrane filters (Jinteng Corp., Tianjin, China). Alkaloids were identified and quantified by comparing retention time and UV absorbance spectra with the standards. In a citric acid-phosphate buffer of pH 3.0, an ion-pair complex was formed between all alkaloids in the sample and the color reagent bromophenol blue upon 5 min of reaction at 30 °C. The complex was extracted with CHCl_3_ in which the absorbance of total alkaloids was measured at the wavelength of 413 nm. Vindoline was used as a reference standard in the preparation of the calibration curve. Each sample was analyzed in triplicate.

### 4.6. Transcriptome Determination and Analysis

#### 4.6.1. Sample Preparation

After pre-culture of *C. roseus* CMC and DDC suspension cultures for 6 day, 25 mg/L *A. flavus* mycelium elicitor was added to the CMCs (the experimental group, EG), and the EG was further cultured for another 48 h. The CMC check (CK) group and DDC group (the callus group, callus) were treated with an equal amount of sterile water.

#### 4.6.2. RNA Extraction, Library Preparation, and Transcriptome Sequencing

The cultures were frozen in liquid nitrogen and ground into powder with a mortar and pestle, and RNA extraction was performed as described by Liu et al. [34]. RNA purity was verified using the NanoPhotometer^®^ spectrophotometer (IMPLEN, Los Angeles, CA, USA). The purity and concentration of the purified RNA were examined with a NanoPhotometer^®^ spectrophotometer (IMPLEN, Los Angeles, CA, USA) and a Qubit^®^ RNA Assay Kit in a Qubit^®^ 2.0 Fluorometer (Life Technologies, Los Angeles, CA, USA).

A total amount of 1.5 μg RNA per sample was used as input material for RNA sample preparation. Sequencing libraries were generated using the NEBNext^®^ UltraTM RNA Library Prep Kit for Illumina^®^ (NEB) following the manufacturer’s recommendations, and index codes were added to attribute sequences in each sample. PCR products were purified (AMPure XP system, Beckman Coulter Inc., Brea, CA, USA), and library quality was assessed on the Agilent Bioanalyzer 2100 system (Genomics & Bioinformatics Core Facility, Fort Wayne, IN, USA).

The clustering of the index-coded samples was performed on a cBot Cluster Generation System using TruSeq PE Cluster Kit v3-cBot-HS (Illumina, Brea, CA, USA) according to the manufacturer’s instructions. After cluster generation, the library preparations were sequenced on an Illumina Hiseq platform, and paired-end reads were generated.

#### 4.6.3. De Novo Transcriptome Assembly and Gene Functional Annotation

Raw data (raw reads) in the FASTQ format were first processed through in-house Perl scripts. In this step, clean data (clean reads) were obtained by removing reads containing adapter and/or poly-N and low-quality reads from raw data.

Transcriptome assembly was accomplished based on the clean data using the Trinity de novo transcriptome assembly software (Trinity Technologies & software Solutions Pvt Ltd., Bengaluru, Karnataka, India) with min_kmer_cov set to 2 by default and all other parameters set to default in absence of a public reference genome for *C. roseus* [35].

All de novo-assembled unigenes were annotated based on the following databases: NCBI non-redundant protein sequences (NR, http://www.ncbi.nlm.nih.gov); NCBI non-redundant nucleotide sequences (NT, http://www.ncbi.nlm.nih.gov); Swiss-Prot (a manually-annotated and reviewed protein sequence database, http://www.expasy.ch/sprot); Protein family (Pfam, http://xfam.org/) [36]; Gene Ontology (GO, http://www.geneontology.org); euKaryotic Ortholog Groups (KOG, http://www.ncbi.nlm.nih.gov) [37], and the Kyoto Encyclopedia of Genes and Genomes (KEGG, http://www.genome.jp/12eg) using BLASTX alignment with an *E*-value = 1e-10 [38,39].

The transcription factor families were identified and annotated using the iTAK software (IAITAM, Canton, OH, USA) [40,41].

#### 4.6.4. Differential Expression Analysis of Unigenes

Gene expression levels were estimated by RSEM [42] for each sample. Clean data were mapped back onto the assembled transcriptome. The read count for each gene was obtained from the mapping results.

Prior to differential gene expression analysis, the read counts were adjusted for each sequenced library using the edgeR program package through one scaling/normalization factor. Differential expression analysis of two samples was performed using the DEGseq R package [43,44]. The *p*-value was adjusted using the *q*-value [45]. *q*-value < 0.005, and |log_2_ (fold change)| > 1 was set as the threshold for significantly differential expression.

GO enrichment analysis of the differentially-expressed genes (DEGs) was implemented by the GOseq R package-based Wallenius non-central hyper-geometric distribution [46], which can adjust for gene length bias in DEGs.

KEGG is a database resource for understanding high-level functions and utilities of biological systems, such as the cell, the organism, and the ecosystem, from molecular-level information, especially large-scale molecular datasets generated by genome sequencing and other high-throughput experimental technologies [47]. KOBAS software (Peking University, Beijing, China) [48] was used to test the statistical enrichment of differential expression genes in KEGG pathways.

### 4.7. Gene Transcription Level Analysis by qRT-PCR

After a 6 days of pre-cultivation of the CMCs, the *A. flavus* mycelium elicitor was added at 25 mg/L. Then, the CMCs (experimental group, EG) and the sterile water-treated CMCs (the check group, CK) were cultured for another 48 h.

The cultures were frozen in liquid nitrogen and ground into a powder with a mortar and pestle, and RNA extraction was performed as described by Liu et al. [34]. The HiScript II 1st Strand cDNA Synthesis Kit (Vazyme, Nanjing, Jiangsu, China) was used to treat 1 μg of total RNA with DNase to remove genomic DNA, after which cDNA was synthesized according to manufacturer instructions.

Transcript levels of the genes encoding the 40S ribosomal protein S9 (*RPS9*, the housekeeping gene), *D4H*, *G10H*, *GES*, *IRS*, *LAMT*, *SGD*, *STR*, *TDC*, and the transcription factor *ORCA3* were monitored in CMC cultures. Primer sequences for *RPS9*, *IRS*, *LAMT* [49], *D4H*, *G10H*, *GES*, *STR*, *TDC*, *ORCA3* [50], *SGD* [27], and *GES* [25] are shown in Appendix A.

The qRT-PCR mixture was prepared using ChamQ SYBR qPCR Master Mix (Q311-02) (Vazyme, Nanjing, Jiangsu, China), and reactions were performed using a LightCycler^®^ 480 II System (Roche, Basel, Switzerland). Amplification included a holding stage of 30 s at 95 °C and 40 cycles, each consisting of 10 s at 95 °C followed by 30 s at 60 °C. Melt curve analysis at 95–60–95 °C was then used to verify the specificity of the amplicons. The expression stability of the qRT-PCR results was assayed using the LightCycler^®^ 480 II Software (Roche, Basel, Switzerland). All samples were measured in triplicate.

## Figures and Tables

**Figure 1 molecules-23-03276-f001:**
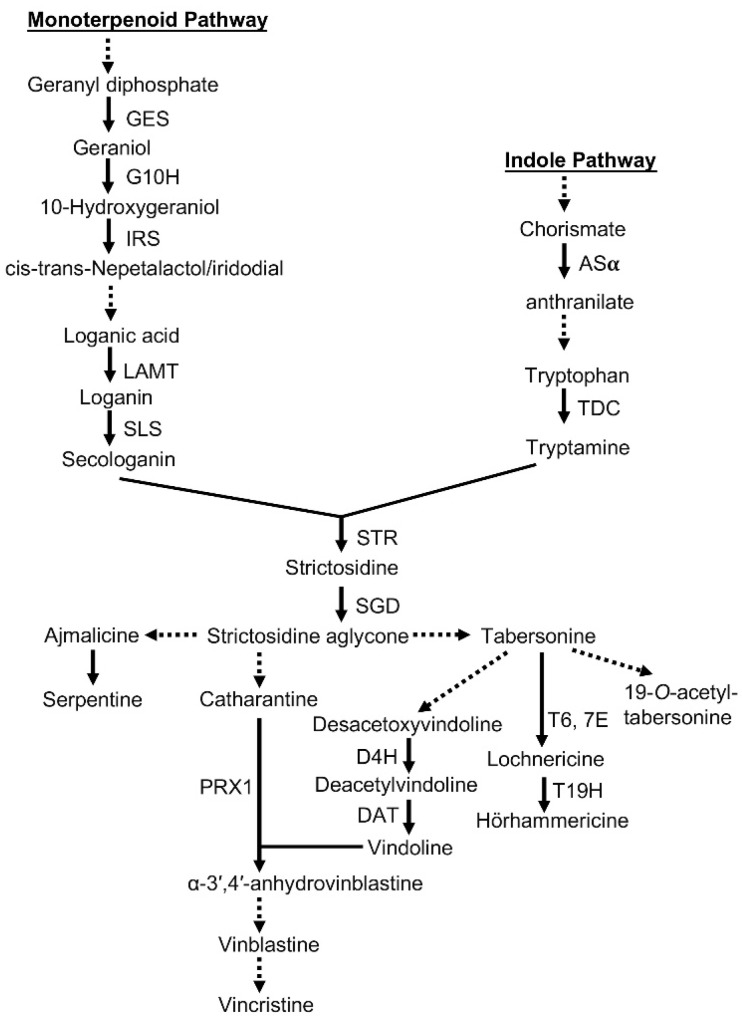
Terpenoid indole alkaloid (TIA) biosynthetic pathways of *C. roseus*. Enzyme abbreviations: GES: geranial synthase; G10H: geraniol 10-hydroxylase; IRS: iridoid synthase; LAMT: loganic acid methyltransferase; SLS: secologanin synthase; AS: anthranilate synthase; TDC: tryptophan decarboxylase; STR: strictosidine synthase; SGD: strictosidine β-d-glucosidase; D4H: deacetoxyvindoline 4-hydroxylase; DAT: deacetylvindoline 4-*O*-acetyltransferase; T6,7E: tabersonine 6,7-epoxidase; T19H: tabersonine/lochnericine 19-hydroxylase; PRX1: vacuolar class III peroxidase. Single arrows denote single steps, and dotted arrows denote multiple or unidentified steps.

**Figure 2 molecules-23-03276-f002:**
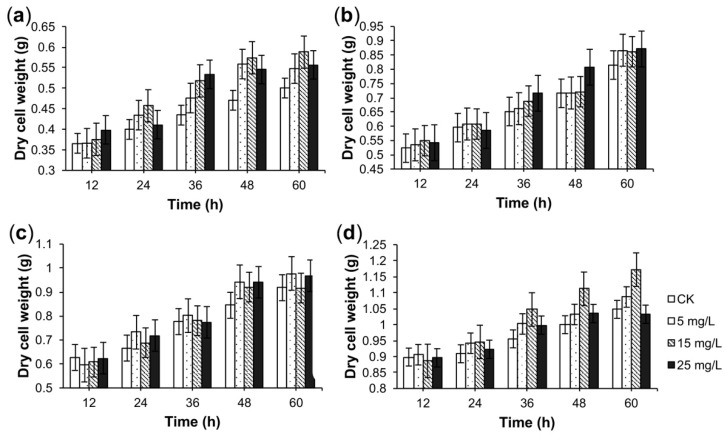
The effect of the *A. flavus* elicitor on the dry cell weight of *C. roseus* cambial meristematic cell (CMC) suspension cultures. The effect of the *A. flavus* medium elicitor on dry cell weight of four-day-old suspensions of *C. roseus* CMCs (**a**) or six-day-old suspensions of *C. roseus* CMCs (**b**). The effect of the *A. flavus* mycelium elicitor on the dry cell weight of four-day-old suspensions of *C. roseus* CMCs (**c**) or six-day-old suspensions of *C. roseus* CMCs (**d**). *C. roseus* CMC cultures treated by sterile water were labeled as the check group (CK). Data are given as the means ± SD (*n* = 3).

**Figure 3 molecules-23-03276-f003:**
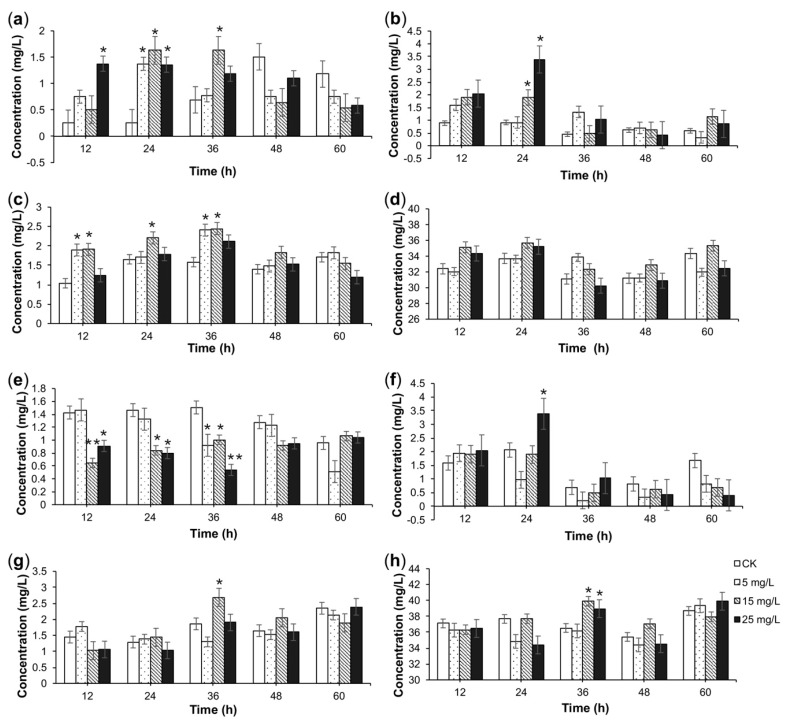
The effect of the *A. flavus* medium elicitor on the concentrations of TIAs. The effect of the *A. flavus* medium elicitor on the concentrations of vindoline (**a**), catharanthine (**b**), ajmaline (**c**), and total alkaloids (**d**) in four-day-old suspensions of *C. roseus* CMCs. The effect of *A. flavus* medium elicitor on the concentrations of vindoline €, catharanthine (**f**), ajmaline (**g**), and total alkaloids (**h**) in six-day-old suspensions of *C. roseus* CMCs. *C. roseus* CMC cultures treated by sterile water were labeled as the check group (CK). Data are given as the means ± SD (*n* = 3). * *p* < 0.05, ** *p* < 0.01, compared to the CK group.

**Figure 4 molecules-23-03276-f004:**
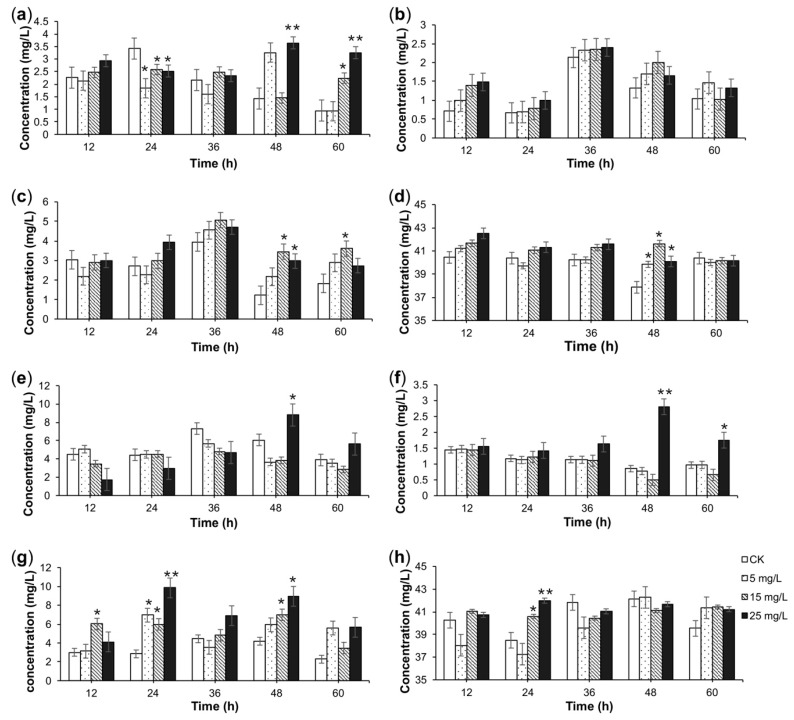
The effects of the *A. flavus* mycelium elicitor on the concentrations of TIAs. The effects of the *A. flavus* mycelium elicitor on the concentrations of vindoline (**a**), catharanthine (**b**), ajmaline (**c**), and total alkaloids (**d**) in four-day-old suspensions of *C. roseus* CMCs. The effect of the *A. flavus* mycelium elicitor on the concentrations of vindoline (**e**), catharanthine (**f**), ajmaline (**g**), and total alkaloids (**h**) in six-day-old suspensions of *C. roseus* CMCs. *C. roseus* CMCs treated by sterile water were labeled as the check group (CK). Data are given as the means ± SD (*n* = 3). * *p* < 0.05, ** *p* < 0.01, compared with the CK group.

**Figure 5 molecules-23-03276-f005:**
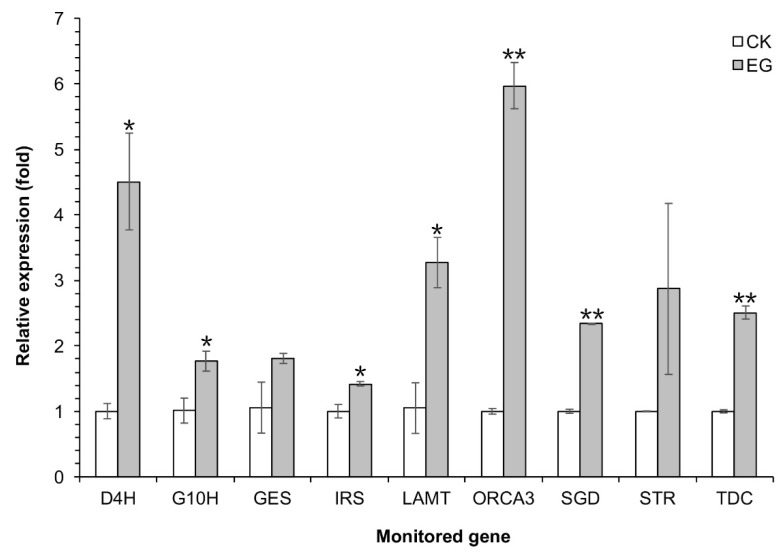
The effects of the *A. flavus* elicitor on the expression of TIA biosynthesis key genes in *C. roseus* CMCs under the optimal *A. flavus* elicitor treatment condition. *C. roseus* CMCs cultured under the optimal *A. flavus* elicitor treatment condition were labeled as the experimental group (EG). *C. roseus* CMCs treated by sterile water were labeled as the check group (CK). Data are given as the means ± SD (*n* = 3). * *p* < 0.05, ** *p* < 0.01, compared with the CK group.

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
