# Peer review of "Effect of Aspergillus flavus Fungal Elicitor on the Production of Terpenoid Indole Alkaloids in Catharanthus roseus Cambial Meristematic Cells"

_molecules, 2018, doi:10.3390/molecules23123276_

Reviewer 1 Report

This manuscript reports the enhanced production of terpenoid indole alkaloids in Catharanthus roseus by the treatment of fungal elicitor from Aspergillus flavus. At first, the manuscript addresses the optimization of treating method of fungal elicitor. Optimized method implements the enhanced production of terpenoid indole alkaloids. The manuscript also shows that the fungal elicitor treatment cause enhanced expression of relevant genes to terpenoid indole alkaloids biosynthesis by RNA-seq and qRT-PCR.

Unfortunately, the fold induction of terpenoid indole alkaloids production is not large. However, the results showed in this manuscript will contribute to increase the production of terpenoid indole alkaloids in Catharanthus roseus. Thus, I think this manuscript can be published after minor revision. I listed several minor comments to be addressed before publication.

1. Chromatograph of HPLC and MS spectra of authentic standards of terpenoid indole alkaloids. These data should be added as supplemental information.

2. The low data of RNA-seq should be deposited to public database, such as SRA in NCBI. This information is useful for other researchers.

Author Response

Point 1: Chromatograph of HPLC and MS spectra of authentic standards of terpenoid indole alkaloids. These data should be added as supplemental information.

Answer: Thanks for reviewer’s kind suggestion. The HPLC-MS/MS analysis is a supporting experiment in this research to prove that there were vindoline, catharanthine and ajmaline in the samples. The peak times of 3 terpenoid indole alkaloids abstracted from samples and their +MS2 spectra were correspond to those of authentic standards in this research. What's more, according to the search results from SCIEX OS 1.3.1 software, their +MS2 spectra have proved what chemicals they were. In addition, as for the presentation of these results, we referred to the report given by Ferreres et al. [1]. Therefore, it is OK that not adding the chromatograph of HPLC and MS spectra of authentic standards. Besides, Figure 3 has been moved to the supporting information part in the revised manuscript.

Point 2: The low data of RNA-seq should be deposited to public database, such as SRA in NCBI. This information is useful for other researchers.

Answer: Thanks for reviewer’s kind remind. We will do further research based on these transcriptome data, so they could not be deposited to public database at present.

Reference

1.     Ferreres, F.; Pereira, D.M.; Valentão, P.; Oliveira, J.M.A.; Faria, J.; Gaspar, L.; Sottomayor, M.; Andrade, P.B. Simple and reproducible HPLC-DAD-ESI-MS/MS analysis of alkaloids in Catharanthus roseus roots. J Pharm Biomed Anal 2010, 51, 65-69, doi:10.1016/j.jpba.2009.08.005.

Reviewer 2 Report

Manuscript No. molecules-384320 entitled “Effect of Aspergillus flavus fungal elicitor on the production of terpenoid indole alkaloids in Catharanthus roseus cambial meristematic cell cultures” provides a new dataset, and these results may be of interest to the readership of the Molecules journal. In my opinion, it may be well cited in the future. Currently, research on biologically active substances is one of the most important branches of biotechnology. Moreover, this type of research shows great utility for the industry. This is proven by the results of these studies. As the authors write A. flavus fungal elicitor can enhance the TIA production of C. roseus CMCs, which can be used as an alternative biotechnological resource for obtaining bioactive alkaloids.

Overall, the manuscript is written in correct scientific language. I did not notice any major factual or editorial errors at work. On the other hand, I do not mention any minor errors, because they are in some sense something normal in this type of work and they do not affect the general perception of it.

In my opinion, the experimental design and data analysis are appropriate, and the introduction and discussion are consistent. Therefore, I have no serious substantive comments to the MS, and in my opinion this work should make a good contribution to the literature.

My conclusion

In my opinion, the work should be accept and published.

Author Response

Point 1: Overall, the manuscript is written in correct scientific language. On the other hand, I do not mention any minor errors, because they are in some sense something normal in this type of work and they do not affect the general perception of it. In my opinion, the experimental design and data analysis are appropriate, and the introduction and discussion are consistent. Therefore, I have no serious substantive comments to the MS, and in my opinion this work should make a good contribution to the literature.

Answer: Thanks for reviewer’s positive comments on our manuscript.

Reviewer 3 Report

In this work, authors report the inducing effect of Aspergillus flavus fungal elicitor on biosynthesis of terpenoid indole alkaloids (TIAs) in Catharanthus roseus cambial meristematic cells (CMCs) under specific culture conditions. Authors relate the increasing accumulations of vindoline, catharanthine and ajmaline with the transcriptome analysis which suggests the increasing expression of numerous genes.

1-     The quantitative analysis of alkaloids was based on HPLC and UV absorbance spectra, and fig 3-(a) shows probably the highest chromatographic peaks obtained in the study (6 days). What are the limits of detection or quantification of the method? This targeted metabolite profile is not very convincing, especially because profile 3-(b) reveals a complex sample with rather low separation quality. Authors should indicate peaks 1,2 and 3 on profile (b), where it is evident that the three TIAs under analysis are not resolved from other interfering analytes present in sample. This limitation seems to compromise the significance of the subtle changes discussed along the whole manuscript. Were the spectra presented on fig 3-(C) obtained with authentic standards?

2-    The comparative quantitative analysis of the three TIAs, essentially presented on figures 4 and 5, reveal that the significant changes are not striking and apparently there is not a clear time- or concentration-dependence of A. flavus elicitor. In addition, for different alkaloids, the induction condition for the highest yield was different. Authors should clarify the criteria for the selection of optimal conditions and discuss in view of these observations, if the n=3 is sufficient to conclusions.

3-    It is also not clear to which specific condition correspond the transcriptome study. The increase on transcript levels of nine genes is correlated with the accumulation of vindoline, catharantine and ajmaline. However, the basis of this correlation is not evident and does not explain per se, the accumulation of intermediates. The mentioned alkaloids are mere intermediates of a complex pathway (figure 1). May authors exclude any downstream blockage and inhibition of gene expression? Did authors considered the possible simultaneous measurement of the end-products of the various pathways branches? To support the authors’ claims, a more comprehensive analysis of TIAs would certainly be more conclusive.

Author Response

Point 1: The quantitative analysis of alkaloids was based on HPLC and UV absorbance spectra, and fig 3-(a) shows probably the highest chromatographic peaks obtained in the study (6 days). What are the limits of detection or quantification of the method? This targeted metabolite profile is not very convincing, especially because profile 3-(b) reveals a complex sample with rather low separation quality. Authors should indicate peaks 1,2 and 3 on profile (b), where it is evident that the three TIAs under analysis are not resolved from other interfering analytes present in sample. This limitation seems to compromise the significance of the subtle changes discussed along the whole manuscript. Were the spectra presented on fig 3-(C) obtained with authentic standards?

Answer: Thanks for reviewer’s thoughtful comments on our manuscript. Peaks 1, 2 and 3 have been indicated on profile (b) according to the reviewer’s suggestion (This figure, titled as Figure S1 in the revised manuscript, has been moved to the supporting information). In order to prove the present of vindoline, catharanthine and ajmaline in the samples, we conducted the HPLC-MS/MS experiment according to the methods of reference [1] with slight modification. The peak times of 3 terpenoid indole alkaloids abstracted from samples and their +MS2 spectra were correspond to those of authentic standards. Therefore, according to the results from SCIEX OS 1.3.1 software, their +MS2 spectra have proved they are vindoline, catharanthine and ajmaline. Besides, in order to conduct the quantitative analysis of alkaloids with good resolution, alkaloid extracts were analyzed by HPLC and ion-pair extraction-spectrophotometry as described by reference [2]. Though this method has some disadvantages, such as HPLC is not extremely sensitive for certain compounds and requires correct operation, this method is effective and practical for the quantitative analysis of alkaloids [3-6].

Point 2: The comparative quantitative analysis of the three TIAs, essentially presented on figures 4 and 5, reveal that the significant changes are not striking and apparently there is not a clear time- or concentration-dependence of A. flavus elicitor. In addition, for different alkaloids, the induction condition for the highest yield was different. Authors should clarify the criteria for the selection of optimal conditions and discuss in view of these observations, if the n=3 is sufficient to conclusions.

Answer: Thanks for reviewer’s thoughtful comments on our manuscript. There were certain changes in the contents of 3 TIAs, though the significant changes were not striking. Overall, the contents of 3 TIAs in C. roseus CMC cultures treated with elicitor presented a trend of first increasing and then reduction with the increasing of time, which were coincidence with those reported [3,7,8]. For certain TIA, there was a concentration-dependence of A. flavus elicitor in this research when other variables remain unchanged. In addition, the results of our preliminary experiment showed that the C. roseus CMC cultures soon turned brown when the dose of A. flavus elicitor was over 25 mg/L. As reviewer suggested, we have clarified the criteria for the selection of optimal conditions-“according to the timing when the contents of vindoline, catharanthine and ajmaline were all relatively high (compared with those of the CK and other experimental groups), optimal condition was confirmed”-in the manuscript. Please see page 6, lines 166-168, in the revised manuscript. Though it is true that the sample size n=3 is small for statistics, statistical analysis is still available when the sample size is n=3. According to previous reports [3,4,8,9], n=3 is commonly used in these experiments. Therefore, the n=3 is sufficient to conclusions in this research.

Point 3: It is also not clear to which specific condition correspond the transcriptome study. The increase on transcript levels of nine genes is correlated with the accumulation of vindoline, catharantine and ajmaline. However, the basis of this correlation is not evident and does not explain per se, the accumulation of intermediates. The mentioned alkaloids are mere intermediates of a complex pathway (figure 1). May authors exclude any downstream blockage and inhibition of gene expression? Did authors considered the possible simultaneous measurement of the end-products of the various pathways branches? To support the authors’ claims, a more comprehensive analysis of TIAs would certainly be more conclusive.

Answer: Thanks for reviewer’s thoughtful and insightful comments. Before we conducted the transcriptome analysis, C. roseus materials were treated with the conditions as follows: after pre-culture of C. roseus CMC and DDC suspension cultures for 6 d, 25 mg/L A. flavus mycelium elicitor was added, and the CMCs (the experimental group, EG) and DDCs (the callus group, callus) were further cultured for another 48 h; the CMC check (CK) group was treated with an equal amount of sterile water (page 11, lines 387-391 in the revised manuscript). Genes were selected from transcriptome data by GO enrichment analysis and KEGG pathway analysis of DEGs, so it is just a coincidence that all selected genes were up-regulated in this research. Among over 130 alkaloids isolated from C. roseus, ajmalicine is a potent antihypertensive reagent [10]; vinblastine and vincristine, two bisindole alkaloids derived from coupling vindoline and catharanthine, are the natural anticancer drugs and are still among the most valuable agents used to treat cancer [11,12]. However, the content of the pharmacological secondary metabolites in C. roseus is very low [13], and commercial production has used a semi-synthetic route to couple catharanthine and vindoline [11]. Thus, in this research, we mainly focused on the contents of vindoline, catharanthine and ajmaline. Overall, though we didn’t determine more alkaloids and genes, our experimental results were sufficient for us to reach the conclusion in this research. Now, we are doing further research based on these transcriptome data, and finding the possible simultaneous measurement of the end-products of the various pathways branches. These comments are very enlightening and helpful to our further research, thank you very much.

References

1.     Ferreres, F.; Pereira, D.M.; Valentão, P.; Oliveira, J.M.A.; Faria, J.; Gaspar, L.; Sottomayor, M.; Andrade, P.B. Simple and reproducible HPLC-DAD-ESI-MS/MS analysis of alkaloids in Catharanthus roseus roots. J Pharm Biomed Anal 2010, 51, 65-69, doi:10.1016/j.jpba.2009.08.005.

2.     Yang, L.; Liu, Y.; Zhang, L.; Zu, Y.G. Ion-pair extraction-spectrophotometric determination of total alkaloids in Catharanthus roseus with bromophenol blue as color reagent. Physical Testing and Chemical Analysis (Part B:Chemical Analysis) 2008, 44, 427-432.)

3.     Zhou, P.; Yang, J.; Zhu, J.; He, S.j.; Zhang, W.j.; Yu, R.m.; Zi, J.c.; Song, L.y.; Huang, X.s. Effects of β-cyclodextrin and methyl jasmonate on the production of vindoline, catharanthine, and ajmalicine in Catharanthus roseus cambial meristematic cell cultures. Appl Microbiol Biotechnol 2015, 99, 7035-7045, doi:10.1007/s00253-015-6651-9.

4.     Wang, M.; Zi, J.; Chen, S.; Wang, P.; Zhu, J.; Song, L.; Yu, R. Artemisinic Acid Serves as a Novel ORCA3 Inducer to Enhance Biosynthesis of Terpenoid Indole Alkaloids in Catharanthus roseus Cambial Meristematic Cells. Natural Product Communications 2016, 11(6), 715-717.

5.     Zhang, Y.; Yuan, Y.; Peng, Y.; Zhu, N. Research on the Growth Dynamic of Medicinal Madagascar Periwinkle. Journal of Anhui Agricultural Sciences 2009, 37(13), 5957-5958, 5960, doi:10.3969/j.issn.0517-6611.2009.13.061.

6.     Yang, L.; Li, J.; Zhao C.; Li, X.; Zu. Y.;Li J. Extraction-Photometric Determination of Total Alkaloids in Camptotheca Acuminata with Methyl Orange as Color Reagent. Physical Testing and Chemical Analysis (Part B:Chemical Analysis) 2009, 45(04), 428-430.]

7.     Zhao, J.; Zhu, W.H.; Hu, Q. Selection of fungal elicitors to increase indole alkaloid accumulation in Catharanthus roseus suspension cell culture. Enzyme Microb. Technol. 2001, 28, 666-672, doi:10.1016/s0141-0229(01)00309-x.

8.     Tonk, D.; Mujib, A.; Maqsood, M.; Ali, M.; Zafar, N. Aspergillus flavus fungus elicitation improves vincristine and vinblastine yield by augmenting callus biomass growth in Catharanthus roseus. Plant Cell, Tissue and Organ Culture (PCTOC) 2016, 126, 291-303, doi:10.1007/s11240-016-0998-1.

9.     Pandey, S.S.; Singh, S.; Babu, C.S.V.; Shanker, K. Fungal endophytes of Catharanthus roseus enhance vindoline content by modulating structural and regulatory genes related to terpenoid indole alkaloid biosynthesis. Scientific reports 2016, 6, 26583, doi:10.1038/srep26583.

10.  Liu, W.; Chen, R.; Chen, M.; Zhang, H.; Peng, M.; Yang, C.; Ming, X.; Lan, X.; Liao, Z. Tryptophan decarboxylase plays an important role in ajmalicine biosynthesis in Rauvolfia verticillata. Planta 2012, 236, 239-250, doi:10.1007/s00425-012-1608-z.

11.  van Der Heijden, R.; Jacobs, D.I.; Snoeijer, W.; Hallard, D.; Verpoorte, R. The Catharanthus alkaloids: pharmacognosy and biotechnology. Current medicinal chemistry 2004, 11, 607-628, doi:10.2174/0929867043455846.

12.  Guo, Z.G.; Liu, Y.; Xing, X.H. Enhanced catharanthine biosynthesis through regulation of cyclooxygenase in the cell suspension culture of Catharanthus roseus (L.) G. Don. PROCESS BIOCHEM 2011, 46, 783-787, doi:10.1016/j.procbio.2010.10.017.

13.  Noble, R.L. The discovery of the vinca alkaloids--chemotherapeutic agents against cancer. Biochemistry and cell biology = Biochimie et biologie cellulaire 1990, 68, 1344-1351.

Reviewer 4 Report

In this manuscript, the authors tried to explore the potential effect of Aspergillus flavus elicitor on the production of three common terpenoid indole alkaloids (ajmaline, catharanthine, and vindoline) and the regulation of some known relevant genes involved in the biosynthesis of TIAs such as D4H, DAT, ORCA3 and so on. However, it is far from qualified for publishing due to its low novelty of methodology and significance of result.  Anyway, I would like to give some suggestions for the authors to modify their data organization especially for the figures and tables which may help them for their submission in future.

1.    It is better to present Figure 2, 4 and 5 in bar chart so as to be visualized more clearly and understood much easier

2.    For Figure 3, I suggest to move panel B and C to supporting information part. What’s more, I think it should be OK to add the corresponding extracted ion chromatograms for the three analytes if you like

3.    Move Figure 6 and 7 to supporting information part as they are more supportive than important

4.    Move all the Tables (1-4) to supporting information part as they are just shown as very basic experimental details but not significant result you should have presented

Author Response

Point 1: It is better to present Figure 2, 4 and 5 in bar chart so as to be visualized more clearly and understood much easier.

Answer: Thanks for reviewer’s kind suggestion. We have presented Figure 2, 4 and 5 in bar chart. Please see pages 4-6, lines 112, 150, 159 in the revised manuscript.

Point 2: For Figure 3, I suggest to move panel B and C to supporting information part. What’s more, I think it should be OK to add the corresponding extracted ion chromatograms for the three analytes if you like.

Answer: Thanks for reviewer’s kind suggestions. Figure 3 has been moved to supporting information part in the revised manuscript (pages 2-3, lines 8-11, in supporting information). In the HPLC-MS/MS analysis, the peak times of 3 terpenoid indole alkaloids abstracted from samples and their +MS2 spectra were correspond to those of authentic standards; what's more, according to the search results from SCIEX OS 1.3.1 software, their +MS2 spectra have proved what chemicals they were. Besides, as for the presentation of these results, we have referred to the report given by Ferreres [1]. So, it is OK that not adding the corresponding extracted ion chromatograms for the three analytes. As reviewer suggested and considering the HPLC-MS/MS analysis is a supporting experiment in this research (to prove that there were vindoline, catharanthine and ajmaline in the samples).

Point 3: Move Figure 6 and 7 to supporting information part as they are more supportive than important.

Answer: Thanks for reviewer’s kind suggestion. As reviewer suggested, Figure 6 and 7 have been moved to supporting information in the revised manuscript (pages 4-5, lines 27-31, 40-43 in supporting information).

Point 4: Move all the Tables (1-4) to supporting information part as they are just shown as very basic experimental details but not significant result you should have presented.

Answer: Thanks for reviewer’s kind suggestion. All the Tables (1-4) have been moved to supporting information part in the revised manuscript according to the reviewer’s suggestion. Please see pages 6-14, lines 51-66, in supporting information part.

Round  2

Reviewer 3 Report

Authors attempted to answer some points of criticism concerning submitted first version. The graphic presentation of quantitative results improved in the revised version. However, the significance of the quantifications remains unclear, which appears to be underestimated when submitted as supplementary results. To demonstrate the clear validation of their own analytical method, authors should clearly present in results section, the fold change of the three TIAs in relation to the limits of detection and quantification for their methodology.

Author Response

Point 1: Authors attempted to answer some points of criticism concerning submitted first version. The graphic presentation of quantitative results improved in the revised version. However, the significance of the quantifications remains unclear, which appears to be underestimated when submitted as supplementary results. To demonstrate the clear validation of their own analytical method, authors should clearly present in results section, the fold change of the three TIAs in relation to the limits of detection and quantification for their methodology.

Answer: Thanks for reviewer’s kind suggestion. HPLC-MS/MS analysis is a supporting experiment in this research (to prove that there were vindoline, catharanthine and ajmaline in the samples). After comprehensive consideration of the reviewers’ suggestions, the figure of HPLC-MS/MS has been move to the supplementary materials. Besides, we conducted the quantitative analysis of alkaloids by HPLC and ion-pair extraction-spectrophotometry as described by Yang [1], which is effective and practical for the quantitative analysis of alkaloids [2-5]. The optimal condition for A. flavus elicitor treatment in C. roseus CMCs was selected by comparing the data as previous reports [2,3,6,7]. As reviewer suggested, we have clarified the method for the selection of optimal condition. Please see page 6, lines 168-170 in the revised manuscript.

References

1.     Yang, L.; Liu, Y.; Zhang, L.; Zu, Y.G. Ion-pair extraction-spectrophotometric determination of total alkaloids in Catharanthus roseus with bromophenol blue as color reagent. Physical Testing and Chemical Analysis (Part B:Chemical Analysis) 2008, 44, 427-432.)

2.     Zhou, P.; Yang, J.; Zhu, J.; He, S.j.; Zhang, W.j.; Yu, R.m.; Zi, J.c.; Song, L.y.; Huang, X.s. Effects of β-cyclodextrin and methyl jasmonate on the production of vindoline, catharanthine, and ajmalicine in Catharanthus roseus cambial meristematic cell cultures. Appl Microbiol Biotechnol 2015, 99, 7035-7045, doi:10.1007/s00253-015-6651-9.

3.     Wang, M.; Zi, J.; Chen, S.; Wang, P.; Zhu, J.; Song, L.; Yu, R. Artemisinic Acid Serves as a Novel ORCA3 Inducer to Enhance Biosynthesis of Terpenoid Indole Alkaloids in Catharanthus roseus Cambial Meristematic Cells. Natural Product Communications 2016, 11(6), 715-717.

4.     Zhang, Y.; Yuan, Y.; Peng, Y.; Zhu, N. Research on the Growth Dynamic of Medicinal Madagascar Periwinkle. Journal of Anhui Agricultural Sciences 2009, 37(13), 5957-5958, 5960, doi:10.3969/j.issn.0517-6611.2009.13.061.

5.     Yang, L.; Li, J.; Zhao C.; Li, X.; Zu. Y.;Li J. Extraction-Photometric Determination of Total Alkaloids in Camptotheca Acuminata with Methyl Orange as Color Reagent. Physical Testing and Chemical Analysis (Part B:Chemical Analysis) 2009, 45(04), 428-430.]

6.     Zhao, J.; Zhu, W.H.; Hu, Q. Selection of fungal elicitors to increase indole alkaloid accumulation in Catharanthus roseus suspension cell culture. Enzyme Microb. Technol. 2001, 28, 666-672, doi:10.1016/s0141-0229(01)00309-x.

7.     Tonk, D.; Mujib, A.; Maqsood, M.; Ali, M.; Zafar, N. Aspergillus flavus fungus elicitation improves vincristine and vinblastine yield by augmenting callus biomass growth in Catharanthus roseus. Plant Cell, Tissue and Organ Culture (PCTOC) 2016, 126, 291-303, doi:10.1007/s11240-016-0998-1.

Reviewer 4 Report

Thanks for the authors' effort on revising the manuscript and the presentation quality has been improved a lot. But based on the result presented in the manuscript, I can not accept it. There is no explicit rule or obvious trend can be identified underlying the elicitation by fungal elicitors on the production of three specific alkaloids.Even under the optimal elicitation condition, the total alkaloids content seems decreased but not increased.  It need to think more about the applicability of the result to other alkaloids, as it somewhat compromised the significance of the research (Refer to Figure 3 and 4). And unfortunately, I can not find any indication that the authors have done biological triplicates for all the experiments. Besides, for the transcriptome analysis, there is not sufficient evidence or data analysis to support the correlation between the up-regulated genes and specific target alkaloid.

Author Response

Point 1: Thanks for the authors' effort on revising the manuscript and the presentation quality has been improved a lot. But based on the result presented in the manuscript, I can not accept it. There is no explicit rule or obvious trend can be identified underlying the elicitation by fungal elicitors on the production of three specific alkaloids. Even under the optimal elicitation condition, the total alkaloids content seems decreased but not increased.  It need to think more about the applicability of the result to other alkaloids, as it somewhat compromised the significance of the research (Refer to Figure 3 and 4). And unfortunately, I can not find any indication that the authors have done biological triplicates for all the experiments. Besides, for the transcriptome analysis, there is not sufficient evidence or data analysis to support the correlation between the up-regulated genes and specific target alkaloid.

Answer: Thanks for reviewer’s thoughtful comments on our manuscript. There were certain trends in the contents of 3 alkaloids, though the significant changes were not striking. Overall, the contents of 3 TIAs in C. roseus CMC cultures treated with elicitor presented a trend of first increasing and then reduction with the extension of time, which were coincidence with those reported [1,2,3]. There are over 130 alkaloids in C. roseus. The biosynthetic pathway of TIAs is complex and highly regulated [4-7]. Different kind of alkaloids response different to the elicitors [2,3,8]. Not all the contents of alkoids would be increased after the cell cultures are treated with elicitor. Therefore, it is common to consider that the total alkaloids content decreased a little after C. roseus CMC cultures treated with elicitor. In this present paper, we focused on three important medicinal alkoids, indoline, catharanthine and ajmaline. Under optimal elicitation condition, the contents of vindoline, catharanthine and ajmaline were 1.45, 3.29, and 2.14 times as high as those of the CK, respectively. So, A. flavus mycelium elicitor indeed have positive effect on the accumulation of the three selective alkaloids.

  We have done biological triplicates for A. flavus elicitation experiment, dry cell weight detection, alkaloid contents detection, transcriptome determination and qRT-PCR experiment. Though our data were not so striking and the sample size n=3 is small for statistics, n=3 is commonly and strictly used in these experiments [1,3,8,9] and statistical analysis is still available when the sample size is n=3.

  Before we conducted the transcriptome analysis, the CMC experimental group was treated with the optimal condition. Genes were selected from transcriptome data by GO enrichment analysis and KEGG pathway analysis of DEGs. What’s more, these genes were all involved in this TIA biosynthetic pathway. The functions of these selected genes in the TIA biosynthetic pathway were showed in Figure 1. Please see page 2, lines 62-69 in the revised manuscript. To further investigate the relation of those genes and the specific target alkaloids, qRT-PCR experiment was done. And it found that the increasing accumulations of vindoline, catharanthine and ajmaline in C. roseus CMCs were correlated with the increasing expression of those genes.

References

1. Zhou, P.; Yang, J.; Zhu, J.; He, S.j.; Zhang, W.j.; Yu, R.m.; Zi, J.c.; Song, L.y.; Huang, X.s. Effects of β-cyclodextrin and methyl jasmonate on the production of vindoline, catharanthine, and ajmalicine in Catharanthus roseus cambial meristematic cell cultures. Appl Microbiol Biotechnol 2015, 99, 7035-7045, doi:10.1007/s00253-015-6651-9.

2. Zhao, J.; Zhu, W.H.; Hu, Q. Selection of fungal elicitors to increase indole alkaloid accumulation in Catharanthus roseus suspension cell culture. Enzyme Microb. Technol. 2001, 28, 666-672, doi:10.1016/s0141-0229(01)00309-x.

3. Tonk, D.; Mujib, A.; Maqsood, M.; Ali, M.; Zafar, N. Aspergillus flavus fungus elicitation improves vincristine and vinblastine yield by augmenting callus biomass growth in Catharanthus roseus. Plant Cell, Tissue and Organ Culture (PCTOC) 2016, 126, 291-303, doi:10.1007/s11240-016-0998-1.

4.  Almagro, L.; Fernándezpérez, F.; Pedreño, M.A. Indole alkaloids from Catharanthus roseus: bioproduction and their effect on human health. Molecules 2015, 20, 2973−3000.

5. Verma, P.; Mathur, A.K.; Srivastava, A.; Mathur, A. Emerging trends in research on spatial and temporal organization of terpenoid indole alkaloid pathway in Catharanthus roseus: a literature update. Protoplasma 2012, 249, 255−268.

6. Peebles, C.A.; Hughes, E.H.; Shanks, J.V.; San, K.Y. Transcriptional response of the terpenoid indole alkaloid pathway to the overexpression of ORCA3 along with jasmonic acid elicitation of Catharanthus roseus hairy roots over time. Metab. Eng. 2009, 11, 76−86.

7. Li, C.Y.; Leopold, A.L.; Sander, G.W.; Shanks, J.V.; Zhao, L.; Gibson, S.I. CrBPF1 overexpression alters transcript levels of terpenoid indole alkaloid biosynthetic and regulatory genes. Front. Plant Sci. 2015, 6, 818.

8. Wang, M.; Zi, J.; Chen, S.; Wang, P.; Zhu, J.; Song, L.; Yu, R. Artemisinic Acid Serves as a Novel ORCA3 Inducer to Enhance Biosynthesis of Terpenoid Indole Alkaloids in Catharanthus roseus Cambial Meristematic Cells. Natural Product Communications 2016, 11(6), 715-717.

9. Pandey, S.S.; Singh, S.; Babu, C.S.V.; Shanker, K. Fungal endophytes of Catharanthus roseus enhance vindoline content by modulating structural and regulatory genes related to terpenoid indole alkaloid biosynthesis. Scientific reports 2016, 6, 26583, doi:10.1038/srep26583.